# A Hemagglutinin Stem Vaccine Designed Rationally by AlphaFold2 Confers Broad Protection against Influenza B Infection

**DOI:** 10.3390/v14061305

**Published:** 2022-06-14

**Authors:** Dian Zeng, Jiabao Xin, Kunyu Yang, Shuxin Guo, Qian Wang, Ying Gao, Huiqing Chen, Jiaqi Ge, Zhen Lu, Limin Zhang, Junyu Chen, Yixin Chen, Ningshao Xia

**Affiliations:** 1State Key Laboratory of Molecular Vaccinology and Molecular Diagnostics, National Institute of Diagnostics and Vaccine Development in Infectious Diseases, School of Life Sciences, Xiamen University, Xiamen 361102, China; zengdian0718@stu.xmu.edu.cn (D.Z.); 21620191152709@xmu.edu.cn (Q.W.); gy36348@163.com (Y.G.); 21620182203347@stu.xmu.edu.cn (H.C.); 21620192203159@stu.xmu.edu.cn (J.G.); 21620180155518@stu.xmu.edu.cn (L.Z.); nsxia@xmu.edu.cn (N.X.); 2State Key Laboratory of Molecular Vaccinology and Molecular Diagnostics, National Institute of Diagnostics and Vaccine Development in Infectious Diseases, Xiamen University, Xiamen 361102, China; xinjiabao@stu.xmu.edu.cn (J.X.); luj04406@xmu.edu.cn (Z.L.); junyuchen@xmu.edu.cn (J.C.); 3Xiamen International Travel Healthcare Center, Xiamen 361012, China; yangkunyu@xmu.edu.cn; 4Faculty of Health Sciences, University of Macau, Macau SAR 999078, China; yc07664@umac.mo

**Keywords:** influenza B viruses (IBV), stem hemagglutinin (HA), vaccine design, structure prediction, AlphaFold2, broad protection

## Abstract

Two lineages of influenza B viruses (IBV) co-circulating in human beings have been posing a significant public health burden worldwide. A substantial number of broadly neutralizing antibodies (bnAbs) have been identified targeting conserved epitopes on hemagglutinin (HA) stem domain, posing great interest for universal influenza vaccine development. Various strategies to design immunogens that selectively present these conserved epitopes are being explored. However, it has been a challenge to retain native conformation of the HA stem region, especially for soluble expression in prokaryotic systems. Here, using a structure prediction tool AlphaFold2, we rationally designed a stable stem antigen “B60-Stem-8071”, an HA stem vaccine derived from B/Brisbane/60/2006 grafted with a CR8071 epitope as a linker. The B60-Stem-8071 exhibited better solubility and more stable expression in the *E. coli* system compared to the naïve HA stem antigen. Immunization with B60-Stem-8071 in mice generated cross-reactive antibodies and protected mice broadly against lethal challenge with Yamagata and Victoria lineages of influenza B virus. Notably, soluble expression of B60-stem-8071 in the *E. coli* system showed the potential to produce the influenza B vaccine in a low-cost way. This study represents a proof of concept for the rational design of HA stem antigen based on structure prediction and analysis.

## 1. Introduction

Influenza viruses are highly infectious respiratory pathogens that cause annual epidemics and periodic global pandemics with relatively high mortality and morbidity [1]. It caused approximately 51,000 deaths and 710,000 hospitalizations in the season of 2017 [2], and averages a USD 11.2 billion economic burden in the United States [3]. Though the majority of influenza cases are caused by seasonal influenza A subtypes H1N1 and H3N2, influenza B viruses caused 25% of all human seasonal influenza infections and are given more attention due to its reassortment with other emerging strains. Influenza B virus has been separated into two main antigenically distinct lineages, Victoria (B/Victoria/2/87-like) and Yamagata (B/Yamagata/16/88-like), since 1983, based on an analysis of the hemagglutinin gene [4]. During the COVID-19 outbreak, the co-infection of SARS-CoV-2 and influenza virus has been common, and patients co-infected with SARS-CoV-2 and influenza B virus have a higher risk of developing poor outcomes [5]. Therefore, there is an urgent demand for developing the universal influenza B vaccine.

Hemagglutinin (HA) is a primary target for neutralizing antibodies and vaccine design [6]. It plays a vital role in the life cycle of the influenza virus by engaging sialic acid receptors on host cell surface to mediate viral attachment, virus–host membrane fusion and infection [7]. Current influenza vaccines provide protection primarily through the induction of neutralizing antibodies against the immunodominant globular head region of the HA which undergoes continuous antigenic drift, and hence, is highly variable among different isolates [6]. Thus, it is critical to distract immune response away from antigenically variable epitopes to subdominant but conserved epitopes for vaccine design. To date, different approaches have been explored based on the immunogenically subdominant HA stem domain. However, these approaches are all based on IAV (influenza A virus) HA stem domain, such as H1, H5 and H3 [8,9,10]. Chimeric HA DNA or protein with a different heterotypic globular head but the same stem region was used to boost stem-specific antibodies after sequential immunization. Trimeric HA stem or HA stem nanoparticles were also explored to generate stem-specific antibodies with protective activity [11]. In addition, shielding the glycan on the variable regions in the HA globular head to redirect the immune responses to the more conserved HA stem region is also used as an approach [12]. However, stem HA vaccine design based on IBV (influenza B virus) has been scarcely reported. Therefore, it is of great significance to extend the strategy of stem HA immunogen design in the development of the universal influenza B vaccine [13,14].

In order to develop a more effective universal vaccine against IBV, we described a novel IBV HA stem vaccine which was conducted by rational design based on structure prediction and analysis of AlphaFold2. B60-stem-8071 vaccine was grafted with an epitope of broadly neutralizing antibodies CR8071 to stabilize the stem HA domain. The results further showed that B60-stem-8071 could generate CR8071-like broadly neutralizing antibodies (bnAbs) and confer robust protection against B/Brisbane/60/2008 and B/Florida/4/2006 in vivo. Structural-based immunogen design has become a novel strategy in the development of a universal IBV vaccine.

## 2. Methods and Materials

### 2.1. HA Stem Sequence Design and Structural Prediction

The HA stem design was based on the HA sequence from B/Brisbane/60/2008 (PDB:4FQM)). To generate the stem HA, the globular head domain was replaced by differently designed linkers, which were the GSA short linker or bnAbs epitopes CR8033 (PDB:4FQL), C12G6, SD84 (PDB:6FYU), 5A7 and CR8071 (PDB:4FQJ), respectively. Next, the removal of the transmembrane and cytoplasmic domains can promote the expression in a soluble form in *E. coli*. Finally, a hexa-histidine tag (his-tag) and a trimerization motif were added at the C terminus for improving the formation of trimer and protein purification. PyMOL Molecular Graphics System, Version1.5.0.3 (Schrödinger, LLC) and ChimeraX 1.3 were used as the molecular visualization software.

All the predicted 3D structures were modelled via AlphaFold2, a deep-learning algorithm which uses protein co-evolution information by multiple sequence alignment (MSA). Inputting the designed sequences, AlphaFold2 would output the predicted structure. In accordance with the pLDDT value given by AlphaFold2 and visual inspection based on empirical knowledge of structural biology, we evaluated the feasibility of whether designed sequences could be folded into stable structures, which helped guide our immunogen design.

ProSA web was used to check the 3D models of the protein structures for potential errors [15].The general quality score evaluated by ProSA for a specific input structure was displayed in a plot showing the scores of all experimentally determined protein chains currently accessible in the Protein Data Bank (PDB): CR8033 (PDB:4FQL), SD84 (PDB:6FYU) and CR8071 (PDB:4FQJ). This feature related to the score of a specific model to the scores computed from all experimental structures was deposited in PDB. Problematic parts of a model were identified by a plot of local quality scores and the same scores were mapped on a display of the 3D structure using color codes.

### 2.2. Cells and Viruses

Madin-Darby Canine Kidney (MDCK) cells were grown in Dulbecco’s Modified Eagles Medium (DMEM, GIBCO,11995-040) containing 10% fetal bovine serum (FBS, GIBCO, Grand Island, NY, USA) and a penicillin–streptomycin mix (100 units/mL of penicillin and 100 μg/mL of streptomycin). Influenza B viruses B/Brisbane/60/2008 (Victoria lineage) and B/Florida/4/2006 (Yamagata lineage) obtained from BEI Resources were grown in MDCK cell at 37 °C for 48–72 h.

Virus was cultured in DMEM containing 1ug/mL TPCK-trypsin and 0.2% Albumin from bovine serum (BSA, GIBCO, Grand Island, NY, USA ). Viruses were inactivated by 0.04% formaldehyde and collected through ultracentrifugation from the supernatant of culture solution. The viruses were then added upon the 30% sucrose cushion and ultracentrifugation at 22,500× *g* for 2 h in 4 °C using a Beckman SW60 rotor. Finally, the supernatant was discarded and virus particles were resuspended in PBS.

### 2.3. Cloning, Expression and Protein Purification

The gene sequences of our design were synthesized (GenScript) and cloned in the expression vector pET-28a (+) (Novagen, San Diego, CA, USA) between *NdeI* (TAKARA, Dalian, China) and *BamHI* (TAKARA, Dalian, China) restriction sites. All constructs were codon-optimized for expression in *E. coli*.

The proteins were expressed in ShuffleT7 *E. coli* (LMAI Bio, Shanghai, China *)* and purified from the soluble fraction of the cell culture lysate. Briefly, a single colony of ShuffleT7 *E. coli* transformed with the designed clone was grown in Luria Bertani (LB) broth overnight at 37 °C until an OD_600_ of about 1.5 was reached. Cells were then induced with 1 mM isopropyl-β-D-thiogalactopyranoside (IPTG, Inalco, Milano, Italy) and grown for another 10 h at 20 °C. Then, the induced cells were harvested at 7000 g, 10 min and resuspended in protein buffer containing 50 mM TB8.0(Sangon Biotech, Shanghai, China) and 50 mM NaCl (SIGMA-ALDRICH, Oakville ON, Canada) with 2% triton-X100 (AMRESCO, Solon, OH, USA). Ultrasonication was used for cell lysis and the supernatant was filtered through a 0.22 μm filter (Millipore, Massachusetts, US). Ni-NTA resin (TransGen Biotech, Beijing, China) was used for protein purification.

### 2.4. Enzyme-Linked Immunosorbent Assay (ELISA)

Overall, 96-well ELISA plates (Wantai BioPharm, Beijing, China) were coated with 100 μL purified viruses (5 μg/mL) or recombinant HA proteins (1 μg/mL) diluted in PBS and incubated for 2 h at 37 °C. The plates were washed once with PBS containing 0.1% *v*/*v* Tween-20 (PBST) and blocked with blocking solution (PBS with 2% sucrose, 0.2% casein-Na and 2% gelatin) for 2 h at 37 °C. Serial 10-fold dilutions of sera or purified antibody were added to the wells and incubated at 37 °C for 30 min. After five washes, 100 μL of horseradish peroxidase (HRP)-conjugated goat anti-mouse (or anti-human) antibody solution was added to each well and incubated at 37 °C for 30 min. After five washes, 100 μL of tetramethylbenzidine (TMB) substrate (Wantai BioPharm, Beijing, China) was added at room temperature in the dark. After 15 min, the reaction was stopped with a 2 M H_2_SO_4_ solution. The absorbance was measured at 450 nm.

For competition ELISA [16,17], it was conducted with an additional preincubation of anti-sera with HRP-conjugated bnAb CR8071 at 37 °C for 2 h, the mixture was then added to the rHA-coated plates and incubated at 37 °C for 2 h. After the washing, TMB substrates were added and stopped with 2M H_2_SO_4_. The OD value was determined at 450 nm, plates were then read on SpectraMax L Microplate Reader (Molecular Devices, 0200-6186). The percent of competition was calculated as follows: % competition = [(A − P)/A] × 100, where A is the signal of CR8071 binding to rHA in the absence of antiserum and P is the binding signal of CR8071 to rHA in the presence of antiserum.

### 2.5. Microneutralization Assay

Microneutralization (MN) assays were conducted as described previously [17]. Briefly, MDCK cells in a 96-well plate were maintained in DMEM supplemented with 10% fetal calf serum (FCS) at 37 °C, 5% CO_2_. Serial 2-fold dilutions of mAbs or sera were mixed with an equal volume of virus and incubated for 2 h at 37 °C. After washing plates two times with PBS, 35 µL of the mixture containing 100 TCID_50_ (50% tissue culture infectious dose) of virus was then added to MDCK cells and incubated for 1 h. The viral supernatant was removed and replaced with DMEM supplemented with 5 μg/mL TPCK-treated trypsin (Sigma-Aldrich, St Louis, MO, USA). The cells were cultured for 48 h at 37 °C in the presence of 5% CO_2_, and the neutralizing titer was determined using the HA test. For the HA test, 50 µL of 0.5% turkey red blood cells was added to 50 µL of cell culture supernatant, and the mixture was incubated at room temperature for 1 h. The serological methods were described previously [18].

### 2.6. Vaccinations and Challenge

Six- to eight-week-old female BALB/c mice were immunized with 80 μg purified B60-Stem-8071 protein with Freund’s complete adjuvant (SIGMA-ALDRICH, MO, USA) as the first immunization and Freund’s incomplete adjuvant (SIGMA-ALDRICH, MO, USA ) as the second and third booster three times at a 2-week interval. Four weeks following the third inoculations, Sera samples were collected and immunized animals were then challenged with B/Brisbane/60/2008 and B/Florida/4/2006 viruses by intranasal inoculation with lethal doses at 1 × 10^5^ TCID_50_. B/Brisbane/60/2008(NR-42005) and B/Florida/4/2006 (NR-41795) were obtained from BEI Resources.

Animals used for T-cell depletion were first injected intravenously with 100 μg anti-CD4^+^ or anti-CD8^+^ antibodies 24 h prior to challenge. The depleted condition was maintained by repeated injections of the monoclonal antibody at 72 h intervals post-challenge. Weights were monitored for 14 days post challenge.

### 2.7. Pathological Analysis

Lung tissues from virus-infected mice were fixed in 10% formalin for at least 24 h. The tissues were then embedded in paraffin by system procedures. Tissue sections were cut and fixed on glass slides, stained by hematoxylin and eosin (H&E).

### 2.8. NK Cell Activation Assay

Sera samples for analysis were firstly treated with receptor-destroying enzyme (RDE) (Denka Seiken, Tokyo, Japan) at 37 °C for 18 h followed by the inactivation of the RDE via incubation at 56 °C for 30 min.

Wells of a 96-well ELISA plate were coated with purified influenza B/Brisbane/60/2008 and B/Florida/4/2006 HA protein (400 ng/well) overnight at 4 °C in PBS. The plates were then washed twice with PBS and 100 µL of sera treated with RDE at 37 °C for 30 min. Plates were then washed five times with PBS and 10^6^ freshly isolated murine PBMCs were added to each well. PBMCs isolated from NC group mice using Ficoll-Paque (GE Healthcare Life Sciences, Uppsala, Sweden ) were washed and resuspended in RF10 media (RPMI 1640, supplemented with 10% FCS, penicillin, streptomycin and l-glutamine; Life Technologies, Grand Island, NY, USA) and were then added to the washed wells and incubated for 5 h at 37 °C with 5% CO_2_. Finally, anti-mouse antibody FITC-antiCD3, PE-antiCD49b and BV421-antiCD107a (Biolegend; used at a 1:400 dilution) were added and incubated at 37 °C for 30 min in the dark. Cells were then fixed with 1% formaldehyde at 37 °C for 10 min and acquisition was performed on the LSRFortessaX-20 flow cytometer (BD Biosciences) with up to10^6^ lymphocyte events collected. Samples were analyzed using FlowJo version 9.2 (Tree Star, Ashland, OR, USA).

### 2.9. Statistical Analyses

All data were plotted and statistical analyses were performed with GraphPad Prism 8.0 software (GraphPad Software, Inc.). Graphs display mean ± standard error of mean (SEM). Statistical significance was analyzed using a two-tailed unpaired Student’s *t*-test. Significance indicated by asterisks is designated as follows: *, *p* < 0.05; **, *p* < 0.01; ***, *p* < 0.001; and n.s., nonsignificant.

## 3. Result

### 3.1. Design of B60-Stem-8071 Vaccine

Although efforts have been made to improve the production of the HA stem domain vaccine, it was still technically challenging to achieve soluble expression in prokaryotic systems since the HA stem domain has not evolved to fold and trimerize as an independent unit [19]. To improve the soluble expression level of the HA stem vaccine, we integrated a rational design and accurate protein structure with the prediction tool AlphaFold2 to reduce the possibility of misfolded protein. Firstly, we designed the vaccine based on the HA stem domain by epitope grafting with different linkers. After that, the predicted 3D structures of our designs would be modelled via AlphaFold2, then the feasibility of whether designed sequences could be folded into stable structures would be evaluated by the energy analysis of ProSA.

Based on the sequence of Victoria representative lineage B/Brisbane/60/2008 (PDB:4FQM) (Figure 1A), the detail of the designed sequences was outlined in Appendix A. Two-stage designs were performed as follows: In the first design phase (phase I, B60-stem), the HA head was omitted from the full-length construct by replacing the deleted fragment by a GSA short linker [20] (Figure 1B). Additionally, we also introduced mutations to further improve the structural stability by referring the design of mini-HA [21]. Mutations were also introduced to mask the hydrophobic patch:A125T, A127T, I128T, C131S and L229S (green), additional mutations (K136C and S256C (orange) ) were to form intermolecular disulfide bonds between monomers to stabilize the HA stem trimer. Mutations were also introduced to destroy HA conformational changes under low pH (L226D and L236D (purple)).The result was shown in a monomer model (Figure 1C). However, the GSA linker(red) failed to connect two discontinuous stem fragments based on the predicted monomer model (Appendix A), thus we speculated that a more flexible peptide was more suitable to link the stem HA domain.

Therefore, in the design phase II, to further improve the stability of the HA stem and enhance the immune response to the epitope of the HA bnAbs, several representative broad neutralization epitopes located in IBV were selected as a flexible linker to stabilize the HA stem domain, CR8033, C12G6, SD84, 46B8 and CR8071 [22,23,24,25,26]. ProSA checked the local model quality and the residue scores were plotted. Negative values suggested no erroneous parts of the model structure [27]. It was predicted that the B60-Stem-8033, B60-Stem-12G6 and B60-Stem-SD84 models were all lying outside the score range of the comparable sized native proteins (Figure 2), indicating poor overall model quality.

Interestingly, B60-stem-8071(−5.71) was predicted as the best qualified model based on the analysis of lower energy by ProSA-web, indicating a more stable conformation, although the Z-score was relatively close to that of B60-stem-5A7(−4.07) (Figure 2). Consistently, a fragment of longer flexible peptide (yellow) can also be observed in the predicted B60-stem-8071 model, suggesting that the CR8071 epitope had more potential to stabilize the stem HA domain in native conformation. Therefore, the design phase I (B60-stem) and the advanced candidate from the design phase II (B60-stem-8071) were compared in the following experiment.

### 3.2. Production and Characterization of B60-Stem-8071

Protein solubility is a coarse indicator of proper folding and remains a crucial problem in prokaryotic expression systems. To investigate whether B60-Stem-8071 can achieve soluble expression in prokaryotic systems, we purified B60-stem and B60-stem-8071 protein from the soluble fraction of the cell culture lysate. The result showed that both of B60-stem and B60-stem-8071 could be purified, but there were still differences in the expression of them (Figure 3A), indicating that grafting the CR8071 epitope linker to the HA stem domain can further promote proper protein folding. The size-exclusion HPLC showed that an additional earlier peak occurred (10 min) for the B60-stem-8071 in non-reduced condition (in red) compared to the reduced one (in black), indicating that the fraction of purified B60-stem-8071 had monomer and trimer (Figure 3B).

Antigenicity of B60-stem-8071 was evaluated by binding activity to CR8071 and CR9114 by ELISA, and the result showed that purified B60-Stem-8071 protein can bind to bnAbs with high affinity (Figure 3C). Additionally, competition ELISA assay showed that anti-B60-Stem-8071 sera inhibited the binding of CR8071 to purified B/Brisbane/60/2008 HA, indicating the CR8071 epitope was grated on the B60-stem-8071 antigen naturally (Figure 3D).

### 3.3. B60-Stem-8071 Broadly Elicits Cross-Reactive Antibodies against Influenza B

To evaluate the broadly protective efficacy of B60-stem-8071 in mice, mice were vaccinated with B60-Stem-8071 (with Freund’s adjuvant). ELISA showed B60-Stem-8071 elicited high titers of cross-reactive antibodies to bind cross-lineage of IBV and early B/Lee/1940 strain (Figure 4A), indicating that the B60-Stem-8071 has adopted a native-like neutral-pH conformation [20]. Microneutralization assay showed that B60-Stem-8071 elicited moderate neutralization against homologous B/Brisbane/60/2008. However, no cross-reactive neutralization against heterologous B/Florida/4/2006 was detected (Figure 4B).

To investigate whether B60-Stem-8071 elicited antibody-dependent cell-mediated cytotoxicity (ADCC) activity, we measured the percentage of CD107a NK cell activation following incubation with purified B/Brisbane/60/2008 and B/Florida/4/2006 HA protein. A significantly higher percentage of NK cell was activated in anti-B60-Stem-8071 sera than NC group (Figure 4C). The detail gating strategy was shown (Appendix A).

### 3.4. B60-Stem-8071 Broadly Protects against Cross-Lineage IBV Challenge

To evaluate the broad protection effect of B60-Stem-8071 against IBV infection in vivo, we accessed the lung tissue damage and weight change after a lethal dose infection of cross-lineage B/Brisbane/60/2008(Victoria lineage) virus and B/Florida/4/2006(Yamagata lineage) virus. (Figure 5A).

Mice vaccinated with B60-Stem-8071 had significantly lower viral titers in the lungs on the third day and the sixth day post-infection compared to the NC group (Figure 5B,C). In addition, histological analysis of lung slices showed obvious inflammatory cell infiltration, alveolar destruction and the thickening of alveolar walls in the NC group. In contrast, for mice vaccinated with B60-Stem-8071, the lung tissue only showed mild pathogenic changes on the third day and the extent of bronchiolitis on the sixth day post-infection (Figure 5F). These results demonstrated that B60-Stem-8071 could reduce the virus load and tissue damage of lungs against cross-lineage IBV infection (Appendix A). Concurrently, mice vaccinated with B60-Stem-8071 continued to have a stable body weight after a lethal dose of viral challenge. Overall, these results demonstrated that B60-Stem-8071 could provide broad protection against IBV infection in vivo (Figure 5D,E).

## 4. Discussion

To date, nearly two-thirds of trivalent inactivated influenza vaccine (TIV) failures were due to the mismatch of influenza B strains [28], which highlights the urgent demand for developing broad-spectrum influenza B vaccines. In this study, we found that cross-protection against IBV infection can be provided by the immunization of our designed B60-Stem-8071 vaccine with Freund’s adjuvant. The HA stem vaccine was grafted with a CR8071 epitope linker, which was rationally screened and engineered based on the structure prediction tool AlphaFold2. It exhibited good solubility and stable expression in the *E. coli* system and had the potential to be applied in large-scale vaccine production. Our study showed that B60-Stem-8071 had potential to become a candidate for broadly protective influenza B vaccine. These results further represented a proof of concept for the rational design of HA stem antigen based on structure prediction and analysis.

The identification of broadly neutralizing epitopes on hemagglutinin (HA) stem domain has raised hopes for developing the universal vaccine providing cross-lineage IBV protection. It was previously reported that HA stem vaccines (H1ssF and H3ssF) presented highly conserved epitopes targeted by human bnAbs and elicited protective responses in mice and ferrets [29], which made a breakthrough in the broad protection against IAV infection. However, there were still many challenges for developing a stem HA vaccine based on short-cycle and low-cost prokaryotic systems. Especially, in prokaryotic systems, how to help proper protein folding and retain the native conformation became the first challenge. Therefore, it was of great importance to design stem HA immunogen rationally. Here, the structure prediction tool AlphaFold2 was used to evaluate our designed models. Since AlphaFold2 can regularly predict protein structures with atomic accuracy, the efficiency of screening our candidates was further improved [30,31].

Next, ProSA-web was applied in the refinement and validation of experimental protein structures and in structural prediction, based on measuring the deviation of the total energy of the structure with respect to an energy distribution derived from random conformations [15]. Similarly, computational design had also been applied in the selection for grafting the backbone HIV gp120 onto an unrelated scaffold protein [32]. In brief, our study further demonstrated the vaccine design approach based on structural biology, which promoted the precise vaccine design targeting the HA stem domain.

However, one of the limitations of our study was that the sample size in vivo experiment was relatively small and thus the data analysis was not that representative enough. Another limitation of our study was that the detailed mechanism(s) by which our stem immunogen provided protection need further investigation. Based on the fact that B60-Stem-8071 can provide cross-protection against IBV infection in mice, we hypothesized that protection against homologous B/Brisbane/60/2008(Victoria lineage) infection was afforded mainly by neutralizing antibodies, while the mechanism of protection against B/Florida/4/2006(Yamagata lineage) was not clear enough. HA stem-reactive antibodies frequently possessed ADCC activity and depended on Fc receptor engagement to confer protection in vivo [33], B60-Stem-8071 can slightly activate a higher percentage of NK cell against cross-lineage IBV HA. This result raised the hypothesis that ADCC might be involved in cross-protection against IBV. Furthermore, the role of T cells was investigated by performing CD4^+^ and CD8^+^ T cell depletion experiment (Appendix A), indicating that CD4^+^ T cells partly engaged in the process, especially during the recovery period. In brief, the cross-protection effect of B60-Stem-8071 was probably through multi-mechanisms. In addition, Freund’s adjuvant was used in our study because it was the most widely used oil–water emulsion adjuvant in animal trials and it stimulated inflammation and facilitated the uptake by antigen-presenting cells (APCs) [34]. Since it was crucial for the selection of adjuvants in the adjuvant-mediated induction of immunity, different types of adjuvants will be evaluated in our following study.

## 5. Conclusions

In summary, soluble expression of B60-stem-8071 in the *E. coli* system indicated that it had the potential to be applied in large-scale vaccine production. The study also represented a proof of concept for the rational design of HA stem antigen based on structure prediction and analysis of AlphaFold2. More importantly, the cross-protection effect of B60-Stem-8071 showed that it can become a candidate for a broadly protective influenza B vaccine.

## Figures and Tables

**Figure 1 viruses-14-01305-f001:**
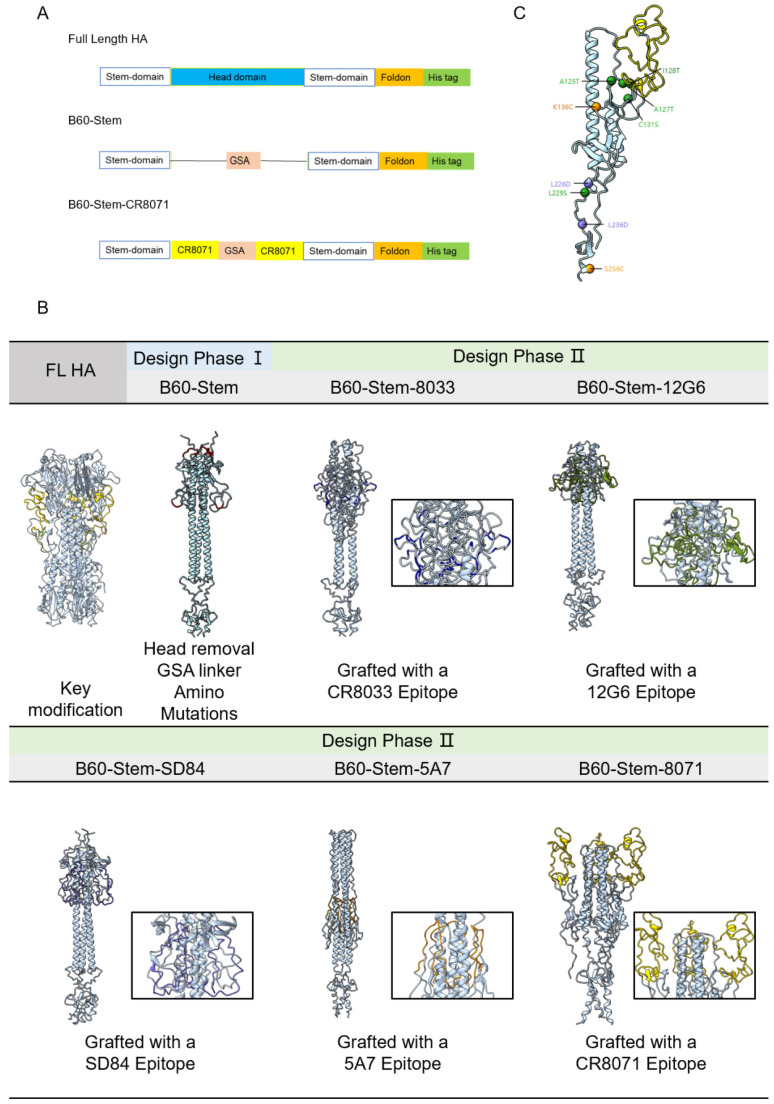
The design of the HA stem strategy and key candidates. (**A**) Schematic of full length, B60-Stem and B60-Stem-CR8071 HA (B/Brisbane/60/2008(Victoria lineage); PDB:4FQM). (**B**) The prediction model of each design phase by AlphaFold2. GSA linker was colored in red, CR8033 epitope (blue), 12G6 epitope (green), SD84 epitope (purple), 5A7 epitope (orange) and CR8071 epitope (yellow) (drawn by PyMOL modeling software). (**C**) Ribbon HA stem model with color-coded modifications.

**Figure 2 viruses-14-01305-f002:**
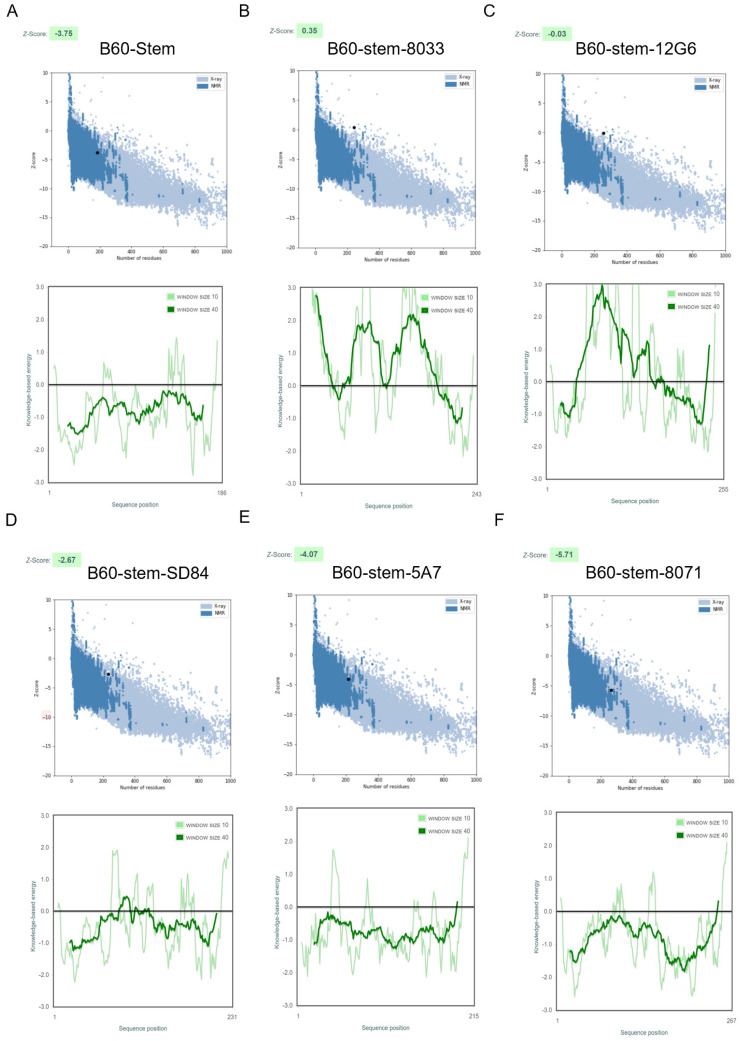
Three-dimensional vaccine design structure validation by ProSA-web. The Z-score of the refined models (**A**) (B60-Stem: −3.75); (**B**) (B60-Stem-8033, 0.35); (**C**) (B60-Stem-12G6: −0.03); (**D**) (B60-Stem-SD84: −2.67); (**E**) (B60-Stem-5A7: −4.07); (**F**) (B60-Stem-8071: −5.71). ProSA-web also plots the residue scores to check the local model quality and the negative values suggest no erroneous parts of the model structure.

**Figure 3 viruses-14-01305-f003:**
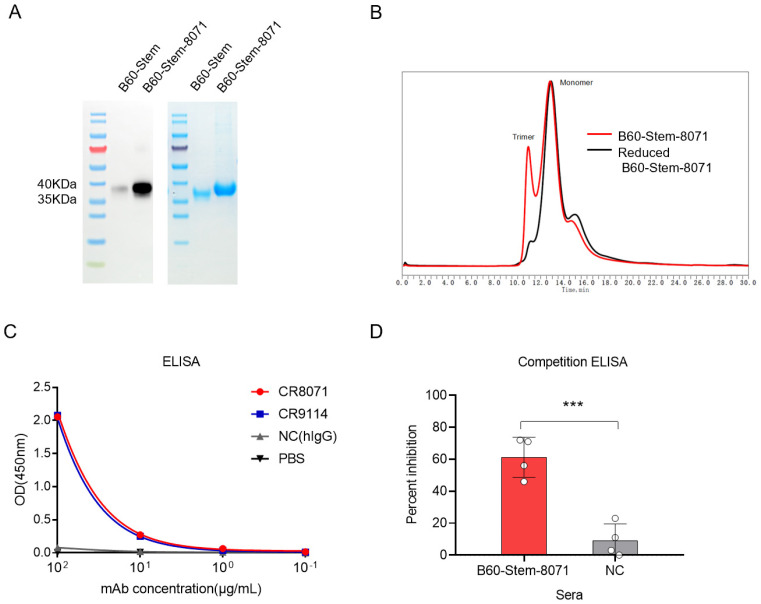
Characterization and conformational determination of B60-Stem-8071 HA. (**A**) The image of SDS-PAGE and the Western blot of purified B60-Stem-8071. (**B**) The HPLC analysis of B60-Stem-8071. B60-Stem-8071 under reducing condition (black) and non-reducing condition (red). (**C**) Binding curves of broadly neutralizing antibodies CR8071 (in red), CR9114 (in blue) and negative (NC) group (non-influenza antibody in grey). ELISA reactivity to purified B60-Stem-8071 (**D**) using a competition ELISA assay with two antisera (B60-stem-8071 and NC). CR8071 was used as a competitor, and NC group mice vaccinated with non-influenza protein. The data represented the average of four independent experiments and error bars indicated the standard errors of the means (SEM), with asterisks representing significant differences (*** *p* < 0.001).

**Figure 4 viruses-14-01305-f004:**
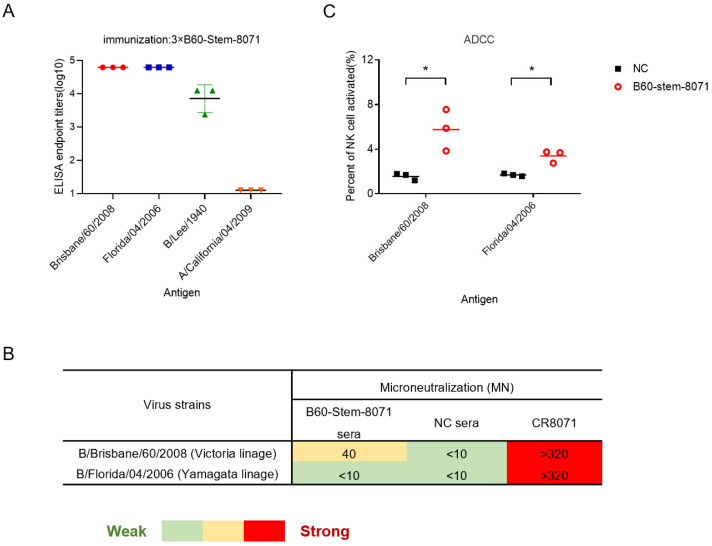
Reactivity of sera from mice vaccinated with B60-stem 8071 against divergent IBV in vitro. (**A**) Broad binding activity of sera from mice vaccinated with B60-stem-8071 against representative inactivated viruses from three influenza B lineages and an IAV pdmH1N1 strain by ELISA. (**B**) Neutralization activity of sera from mice vaccinated with B60-stem-8071 or adjuvant (NC group) to IBV by MN assay. The neutralization strength is color coded from green (weak) to red (strong). The value <10: no reactivity (filled with green); 10–40: moderate reactivity (filled with yellow); >40: high reactivity (filled with red). Asterisks represented significant differences Significance (* *p* < 0.05). (**C**)The ADCC effect of sera from mice vaccinated with B60-stem-8071 or NC group.

**Figure 5 viruses-14-01305-f005:**
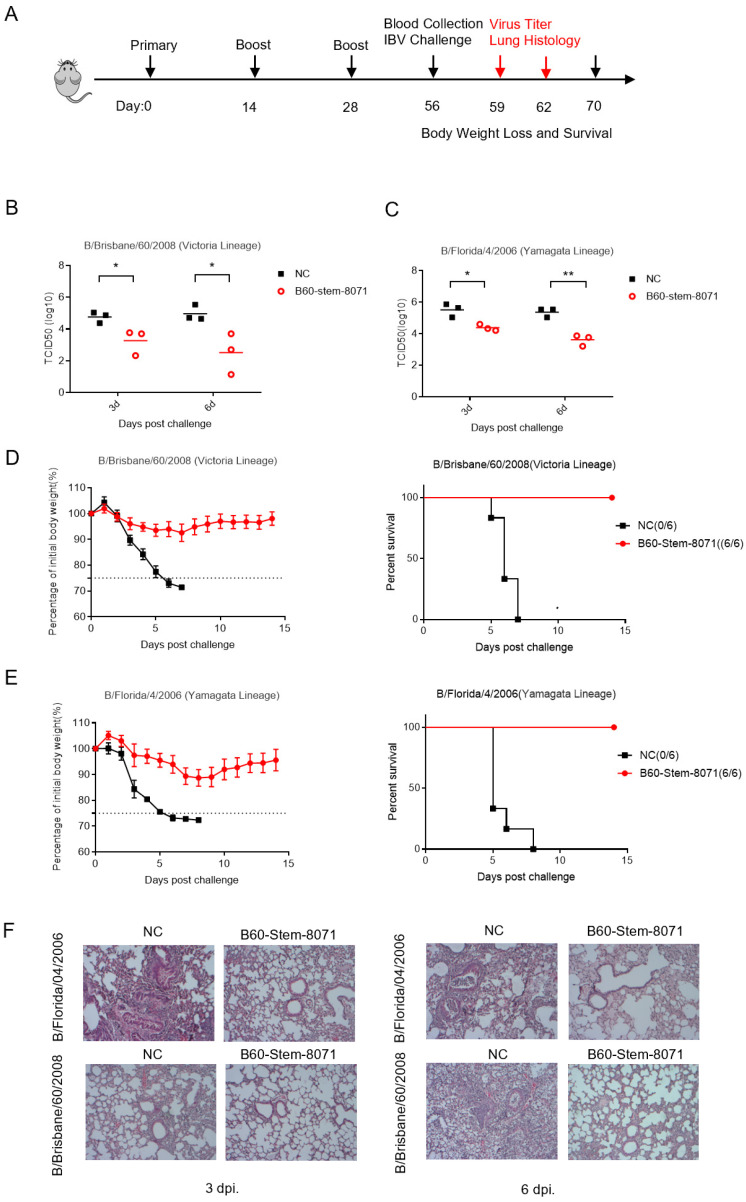
B60-Stem-8071 conferred robust broad protection against cross-lineage IBV infection in vivo. (**A**) Diagram showed experimental plan. Twelve mice per group vaccinated with B60-Stem-8071 or adjuvant (NC group) were infected with B/Brisbane/60/2008(Victoria lineage) or B/Florida/4/2006(Yamagata lineage) intranasally. Three mice per group were euthanized on the third and the sixth day post-infection. Bronchoalveolar lavage fluid was collected to determine viral titer, and lungs were harvested for histopathology. (**B**,**C**) Graph showed mean lung virus titer ±SEM (*n* = 3 mice) on days 3 and 6 post-infection, with asterisks representing significant differences Significance (* *p* < 0.05, ** *p* < 0.01). (**D**,**E**) Mouse body weight curves ±SEM (*n* = 6 mice) and survival curves. (**F**) The H&E staining of lung sections harvested from B60-Stem-8071 or NC group.

## Data Availability

The data presented in this study will be available on request from the corresponding author.

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
