# Peer review of "A Hemagglutinin Stem Vaccine Designed Rationally by AlphaFold2 Confers Broad Protection against Influenza B Infection"

_viruses, 2022, doi:10.3390/v14061305_

Round 1

Reviewer 1 Report

This ms describes the design and preliminary testing in vitro and in vivo of a ‘universal’ influenza B vaccine directed against the hemagglutinin stem.  This is potentially of great interest; the results presented are promising.  However, the ms needs a lot of work to make it publishable.

Regarding the data itself, this reviewer is not qualified to comment on the structural predictions and validations made using AlphaFold2 and ProSA programs, with which I am not familiar.  Also I was unable to access the supplementary figures.  The western blot, antibody binding, neutralization, and mouse survival studies appear to be competently done.  Some issues are:

1) Materials and Methods line 160 states that mice were immunized 3 times, but everywhere else it appears they were immunized 2 times—which is it? 

2) Lines 161-162, how much virus was used for the challenges, in pfu or TCID50 units? 

3) Lines 164-167, T-cell depletion experiments are mentioned here and at the end of the Discussion (lines 348-350) but nothing is said about these experiments in Results.  Were these experiments actually done? 

4) NK cell activation assay:  Please provide a reference for this protocol. Line 173—is this 400 ng per well?  Line 175, ‘blocked with RPMI1640 medium’—I suspect that blocked is the wrong term as RPMI1640 is not a blocking agent and I don’t understand why the wells should be blocked.  Line 176, suggest insert ‘incubated with’ before ‘100’.  What is the source of the PBMCs?  Are they murine PBMCs?  What positive control was used? The preparation of the NK cells is best described at the beginning, not the middle of the paragraph. Line 179, ‘cells’ should probably be wells.  Please define ADCC.  The authors might consider showing in Results the scatterplot graphs of the flow cytometry output.

5) Figure 1B would be easier to interpret if it were about twice as large.

6) Figure 3B, the color coding in the figure itself is the reverse of that described in the figure legend. For Figure 3D, the legend is confusing—is this “one representative data set” or is this the average of the 4 repetitions?

7) The English usage demands a thorough editorial going-over.  The Introduction is not bad but thereafter there are numerous problems.  Some repeated ones are writing ‘stable’ for stabilize, ‘gratify’ or ‘grift’ for grafting, capitalization in the wrong places, absence of a needed preposition.  Some specific issues noted were: Line 80, ‘trimer’ should be trimerization; line 110, ‘rotate’ should be rotated; line 119, ‘designed’ should possibly be desired I think; line 128, ‘2 hours coating in’ should be 2 hours at; line 261, ‘linage’ should be lineage; line 294, ‘wight’ should be weight.  The abbreviation ‘bnAbs’ is not defined and I take it to mean broadly neutralizing antibodies.  In line 154, I suppose that TRBC means turkey RBCs and the meaning should be made explicit.

Author Response

Dear Reviewer:

Thanks for all your comments concerning our manuscript.

These comments are all valuable and  helpful for revising and improving our paper. We have carefully considered the comments and have revised the manuscript accordingly. We hope the revised manuscript will meet with approval.

Please see the attachment.Thank you!

Reviewer 2 Report

The work by Zeng et al. reports the generation of a new vaccine prototype based on a subdominant and conserved epitope from the HA of an IBV strain. This prototype was predicted through in silico approaches, considering rational design, structure prediction and analyses, and different stabilization mechanisms for this antigen, mostly using the AlphaFold2 AI system. It was then successfully expressed in a prokaryotic system. They also evaluate the protective capacity of this vaccine prototype upon influenza virus infection of mice. NK cell activation assays are also performed, suggesting a possible ADCC induction by the antibodies generated upon immunization of mice with this prototype. The results are promising and could lead to further evaluation of this vaccine prototype in different preclinical models.

General comments

This article is relevant and interesting, showing data for a new vaccine prototype, both as a proof of concept and in in vivo experiments. The conclusions drawn in the manuscript are clear, coherent, and mostly well supported by the data obtained. The controls included are adequate to reach the corresponding conclusions. However, the manuscript would greatly benefit from a new independent experiment for the in vivo results.

The quantity and the quality of the references cited throughout the manuscript are good overall. Most of them are current (from the last five years) and related to the topic covered in the manuscript.

The figures included in the manuscript are mostly clear and relevant to show the data obtained.

In some places, there are references to experimental data not shown (i.e., T cell depletion), and this should be addressed.

No information regarding the statistical analyses performed is included anywhere in the manuscript, and this should be addressed.

No information regarding ethical approvals is indicated throughout the article, and this should be addressed as experiments with mice are performed.

No informed consent is required in this article. The authors report no conflicts of interest. Funding (acknowledgments) is clearly indicated.

The English grammar and general language require major and thorough revisions. Also, a thorough typesetting revision should be made (some comments are indicated below). You could use grammar revision software to help you with this (such as Grammarly or Ginger).

Specific comments:

A thorough English use, grammar, and typesetting revision of this manuscript should be performed.

Line 37: Is this number referring to a worldwide burden? Or specific to one country?

Line 70: Define bnAb here (it was defined in the abstract but should be defined again in the main text).

Line 78: This sentence is confusing and requires clarification.

Line 82: Are these colors referring to Figure 1? If so, this figure needs to be cited here.

Line 106: Are these viruses from ATCC or BEI resources? If so, please indicate it. Otherwise, indicate their origin (i.e., cite previous publications).

Line 110: Did you mean that the virus was incubated in an orbital shaker? For how long?

Line 126: Indicate in the title what these ELISAs were for.

Line 127: Please rephrase this.

Line 128: Which buffer did you use to coat the plates?

Line 131: Is this also PBS? Please keep using the acronym if so.

Line 139: Please clarify the competition ELISA methodology, as it is not clear (i.e., which monoclonal antibodies you used?).

Line 158: A figure (could be supplemental) showing this challenge and immunization setup should be included. Otherwise, you could update Figure 5 to include the immunization times and not only the challenge and euthanasia times.

Line 159: Which protein?

Line 162: Please indicate the catalog number for the BEI-obtained viruses.

Line 164: There is no data regarding this methodology in the results section.

Line 173: Is this "purified" and inactivated virus?

Line 174: Why did you use PBS here and not carbonate/bicarbonate buffer?

Line 175: Why did you block with medium? Was this supplemented with FBS? Why did you not use the same blocking solution as before?

Line 176: What does RDE stand for?

Line 177: Where were these cells purified from? Mice or healthy donors?

Line 179: If these cells are the same ones indicated above (line 177), this sentence should be before adding the cells to the plate. If they are different NK cells, you should indicate whether they are mouse or human NK cells.

Line 180: Since you are using mouse antibodies, I assume the first NK cells mentioned originated from mice. Either way, please state it more clearly.

Line 182: Regarding formaldehyde, for how long? What temperature? Why again?

Line 192: What did you mean by gratifying?

Line 197: There was no supplementary material available on the platform, so I could not check this.

Line 209: Since these Z-scores are included in Figure 2, you can cite this figure and reduce this sentence just to indicate that a bad overall model quality was detected for these models.

Line 230: "The Z-score…" is shown?

Line 239: There is no quantification of this figure. If you want to state this soundly, please include a quantification of the bands on these images (densitometry works just fine).

Line 256: Why is only one data set shown here? Are the results too different? I would suggest including all the data obtained. Which statistical test was performed to obtain a p-value here?

Figure 4: Did you perform statistical analyses for this figure? If so, please indicate them in the legend.

Figure 4A: You should also refer to the IAV strain evaluated in this figure.

Figure 4B: What is the meaning of the numbers on this table? Are they representing dilution factors where a certain percentage of the virus is neutralized? This should be explained, either in the figure legend or the main text.

Figure 4C: Could you include a gating strategy for this assay in the supplementary material? Is this percentage relative to parental gating? Also, did you include absolute count beads in these assays? If you did not, could you indicate the percentage of total NK cells in the previous gating and normalize your percentage of activated NK cells to this value? This reduces variability in the percentages obtained when parental gatings are not homogeneous.

Line 286: Please use euthanized rather than sacrificed.

Line 286: This sentence is confusing, as it suggests that you harvested lungs from the same mouse on days 3 and 6 while performing euthanasia on day 14. To my understanding, you performed euthanasia of these mice on days 3, 6, and 14. Please rephrase this, so it is more precise.

Line 291: "of…" immunized mice?

Line 294: Please rephrase.

Figures 5B and C: Did you perform statistical analyses here?

Figure 5F: Could you include a histopathological quantification for the H&E staining performed?

Line 314: Why did you decide to use Freund's adjuvant? Was it because it promotes a Th1 polarization of T cells and high titers of antibodies? Why did you not evaluate other adjuvants already approved for human use? Please refer to this in your discussion.

Line 323 and 334: Include et al. here because this was a work performed by several authors and not only Nicole or Mihai.

Line 339: You should also mention here that the sample size of your in vivo experiment was relatively small (3 for each time point), and only a single independent experiment was performed.

Line 348: This is nowhere in the results. It is also mentioned in the methodology, but no results regarding T cell depletion are shown. Please include this data or delete this statement here and the methodology associated.

Please indicate the catalog number or further information for the following reagents and equipment:

Line 100: PDB accession number.

Line 103: ATCC catalog number.

Line 116: Catalog numbers for the expression vector and the restriction enzymes used.

Line 138: ELISA reader's name and catalog number.

Line 118: Did you mean Shuffle? If so, please also indicate the catalog number.

Line 121: Catalog number for IPTG and the reagents used in this section.

Line 160: Catalog number for the adjuvant.

Line 183: Is this an LSRFortessa? Plain or X-20?

Author Response

Dear Reviewer:

Thanks for all your comments concerning our manuscript.

These comments are all valuable and helpful for revising and improving our paper. We have carefully considered the comments and have revised the manuscript accordingly. We hope the revised manuscript will meet with approval. 

Round 2

Reviewer 1 Report

Overall my previous criticisms have been satisfactorily addressed.  I have no further comments on the science. There remain a few places where the English needs attention (capitalization issues in lines 170, 334, 574, 754, 808; syntax in lines 188-189, 203, 741, 765; "106 in line 335 should probably be 106 . In lines 553-554 "undetectable...was detected"  would be better as "no ...was detected.")

Author Response

Dear Reviewer:

Thanks for your time involved in reviewing the manuscript and your encouraging comments on the merits.

We also appreciate your clear and detailed feedback and hope you will find this revised version satisfactory, the responses to your comments are as follows.

Best regards

sincerely,

Yixin Chen

Reviewer 2 Report

The authors have addressed all my comments and concerns and I believe that the manuscript is in good shape to be published. Thank you for the opportunity to review this interesting article.

I wish you the best.

Author Response

Dear Reviewer:

Thanks for your time involved in reviewing the manuscript and your encouraging comments on the merits.

We also appreciate your clear and detailed feedback and hope you will find this revised version satisfactory.

Best regards

sincerely,

Yixin Chen
